

# Magnitude of non-adherence to antiretroviral therapy and associated factors among adult people living with HIV/AIDS in Benishangul-Gumuz Regional State, Ethiopia

Fikadu Tadesse Nigusso[1,2] and Azwihangwisi Helen Mavhandu-Mudzusi[1]

[1] Department of Health Studies, University of South Africa (UNISA), Pretoria, South Africa
[2] Nutrition and Education Section, United Nations World Food Programme (WFP), Addis Ababa, Ethiopia

Corresponding author
Fikadu Tadesse Nigusso,
fike1f@gmail.com

## ABSTRACT

**Introduction.** Following global efforts to increase antiretroviral therapy (ART) access and coverage, Ethiopia has made significant achievement with a 6.3% annual decline in the HIV/AIDS incidence rate between 1990 and 2016. Such success depends not only on access to ART but also on attaining optimum treatment adherence. Emerging studies in Ethiopia has shown the increasing prevalence of poor adherence and lack of the desired viral suppression, but the extent and factors associated with non-adherence to ART are not well known, especially in the current study setup. In this study, we examined the magnitude and factors associated with treatment and non-adherence to ART among people living with HIV in Benishangul-Gumuz Regional State, northwest Ethiopia.
**Methods.** An institutional facility based cross-sectional descriptive study was carried out among adult people living with HIV/AIDS from mid-December 2016 to February 2017 with only 98.9% response rate. Sociodemographic factors (age, gender, marital status and residential area), economic factors (educational status, income, asset possession, employment status, dietary diversity, nutritional status and food security), and clinical characteristics (CD4 count, duration on ART and history of opportunistic infections) were explanatory variables. ART non-adherence was measured using a visual analogue scale (VAS). We used binary logistic regression and subsequent multivariate logistic regression analysis to determine the factors associated with ART non-adherence.
**Result.** Overall, 39.7% of the participants were found non-adherent to ART. Strong association was found between non-adherence to ART and young age below 25 years (AOR: 4.30, 95% CI [1.39–3.35]; $p = 0.011$), urban residential area (AOR: 2.78, CI [1.23–7.09], $p = 0.043$), lack of employment (AOR: 1.75, 95% CI [1.05–2.91], $p = 0.032$), food insecurity (AOR: 2.67, 95% CI [7.59–8.97]; $p < 0.0001$), malnutrition (AOR: 1.55, 95% CI [1.94–2.56]; $p = 0.027$) and opportunistic infections (AOR: 1.81, 95% CI [1.11–2.97]; $p = 0.018$).
**Conclusion.** The prevalence of non-adherence to ART in this study was high. Sociodemographic and economic factors such as young age of below 25 years, urban residential area, lack of employment, food insecurity, malnutrition and opportunistic infections were among the factors associated with non-adherence to ART.

## INTRODUCTION

Antiretroviral therapy (ART) has played a significant role in responding to HIV/AIDS epidemics (*Granich et al., 2012*; *Williams, Lima & Gouws, 2011*). It has increased survival and improved quality of life, and reduced the rate of disease progression and death (*Oguntibeju, 2012*; *Dalhatu et al., 2016*). Consequently, ART service expansion has received significant support at the global and national level through the commitment of the Sustainable Development Goals (SDG) (*UNAIDS, 2015*; *The Federal Democratic Republic of Ethiopia Ministry of Health, 2015*). The SDG response sets out targets for HIV treatment: 90% of people living with HIV know their HIV status, 90% of people who know their status receiving treatment and 90% of people on treatment having a suppressed viral load (*UNAIDS, 2011*). In line with this initiative, HIV treatment and care services have expanded dramatically (*UNAIDS, 2018*). In 2017, over 21.7 million people living with HIV were receiving ART globally (*UNAIDS, 2018*). Regardless of this increase, ensuring adherence to HIV treatment remains challenging for the HIV/AIDS response, mostly in sub-Saharan Africa (*Fonsah et al., 2017*; *Eyassu, Mothiba & Mbambo-Kekana, 2016*; *Wang et al., 2011*) including Ethiopia (*Alagaw et al., 2013*; *Peltzer & Pengpid, 2013*).

Scientific literature indicates that for ART to be effective and prevent the emergence of resistant strains, a strict adherence level of ≥95% is recommended. This was based on Paterson's pioneer study that found up to 95% adherence is necessary for effective HIV viral suppression (*Paterson et al., 2000*). Once initiated, ART must be continued daily (*World Health Organization, 2010*). However, irrespective of the benefits of good adherence, not all people living with HIV/AIDS (PLWHA) fully adhere to the required treatment level. Failure to attain the required adherence level results in poorer prognosis, higher morbidity, mortality and the development of resistance to ART (*Iacob, Iacob & Jugulete, 2017*; *Nachega et al., 2011*). Once resistance develops, it has dire consequences for individual PLWHA, family, community, healthcare providers and the healthcare system because treating ART resistant HIV strain requires second-line ART regimens, that are more expensive and have worse side effects (*Chauhan et al., 2019*; *Karade et al., 2018*).

Given the importance of adherence, a growing body of research has identified factors associated with non-adherence to ART as behaviors related to the patient itself, therapy related (medication regimens such as more than one tablet or dosing time per day and ART toxicity), poor relationships with healthcare providers and socioeconomic factors (*Kim et al., 2018*; *Mohammed, Ahmed & Tefera, 2015*; *Wasti et al., 2012*). Poor ART adherence was reported among underprivileged communities where socioeconomic inequalities heighten the risk (*Young et al., 2014*) and lower quality of life is prevalent (*Liping et al., 2015*; *Hansana et al., 2013*), negatively impacting the HIV program (*Wang et al., 2011*). For PLWHA, factors related to gender dimensions, socioeconomic factors such as education status, income, asset possession and food security have been associated with non-adherence to ART (*Basti et al., 2017*; *Abera et al., 2015*; *Alagaw et al., 2013*). For example, in South

Africa (*Eyassu, Mothiba & Mbambo-Kekana, 2016*), Cameroon (*Fonsah et al., 2017*) and northern Tanzania (*Samuel Edward et al., 2018*), gender, income and level of education were reported as the determinants of ART adherence. Another study in Wolaita Sodo, Ethiopia, found lack of food was associated with poor adherence (*Alagaw et al., 2013*). In opposition to the culminated evidence, there are studies that have reported an absence of the impacts of socioeconomic inequalities on ART adherence in middle and low-income countries (*Peltzer & Pengpid, 2013*).

Ethiopia is among sub-Saharan African countries sturdily hit by epidemics. The country has made a significant achievement in responding to the HIV/AIDS epidemic, with a 6.3% annual decline in the HIV/AIDS incidence rate between 1990 and 2016 and a total reduction of 77% (*Deribew et al., 2019*). Regardless of the government's commitment to end the HIV/AIDS epidemic as per the 2030 SDG initiative (particularly one that envisages a 90% suppression of viral loads to prevent treatment failure and reduce AIDS-related deaths), achieving the required level of suppressed viral load is becoming a challenge in Ethiopia. Current evidence in the country has shown that only 32% of PLWHA on ART have suppressed viral loads (*UNAIDS, 2017*). Studies have indicated that lack of viral suppression is an indication of non-adherence to ART (*Byrd et al., 2019*; *Nasuuna et al., 2018*; *Robbins et al., 2014*). Therefore, understanding the level of adherence and factors related with non-adherence helps to prevent the future threat of widespread treatment resistant HIV strains. We, therefore, aimed at investigating the magnitude and associated factors with non-adherence to ART among adult people living with HIV and on treatment in the Benishangul-Gumuz Regional State (hereafter Benishangul-Gumuz) in northwest Ethiopia. Our results will inform implementation of strategic programs and practices to improve ART adherence to achieve optimal HIV care in the country.

## METHODS
### Study design and population
A cross-sectional study was conducted from mid-December 2016 to February 2017 in Benishangul-Gumuz, one of the nine regional states of the Federal Democratic Republic of Ethiopia located in the northwestern part of the country. The study was conducted among two referral hospitals and three health centers which provide comprehensive HIV care services. These health facilities were purposely targeted as they are utilized by the majority of people living with HIV. During this study period, there were a total of 2,721 PLWHA who were on ART treatment follow-up among the selected facilities. Sample size for this study was determined using the formula for the estimation of single proportion, $n = \frac{(z)^2 p(1-p)}{d^2}$, where proportion (p) of 63% taken from the previous study in Ethiopia (*Tiyou et al., 2012*) margin of error (d) = 5%, and 95% confidence limit ($Z = 1.96$). By adding 10% to cater for non-response rate, a total of 394 respondents were enrolled into the study. This was allotted to the study sites proportionally to the number of case load at each facility. Finally, a simple random sampling technique using a sampling frame developed from the registration book of the patients was used to enrol respondents daily at each study sites. The study entry criteria were: (a) PLWHA who had received at least
**Table 1  Study sites and sample contribution.**

| Name of health facility | Accessible population (M) | Samples (frequency, %) $n = \sum[(\frac{M}{N}) * 394]$ |
|---|---|---|
| Hospital A | 1,079 | 156 (40%) |
| Hospital B | 921 | 133 (34) |
| Health Centre X | 275 | 40 (10) |
| Health Centre Y | 251 | 37 (9%) |
| Health centre Z | 195 | 28 (7%) |
| Total (N) | 2,721 | 394 (100%) |

**Notes.**
M, the number of peoples living with HIV and AIDS receiving ART at each health facilities; N, total number of PLWHA receiving ART at the study sites during the study period; n, number of samples proportionally drawn from each study site.

one-month antiretroviral therapy and on follow-up at the selected health facilities, and (b) resided in Benishangul Gumuz Regional State for at least two years, (c) signed consent to participate in the study. Exclusion criteria: (a) below 18 years of age, (b) critically ill, (c) psychiatric health problem as previously diagnosed and confirmed. Table 1 presents the number of accessible populations or PLWHA receiving ART during the study period and sample contribution among the study sites.

## Data and measures

A structured questionnaire was used for face-to-face interviews with the study population during times participants were waiting to see ART service providers and for pharmacy refills. The study tool was checked for clarity and consistency among the selected health facilities out of the study sites by a group of health providers over a two-week period. The result assisted in ensuring the quality and consistency of the instrument. Training was given to data collectors about the objectives of the study, and how they could approach the patients to obtain their confidentiality. The collected data kept in locked cabinets and password protected computers during data processing and analysis. Data extraction forms were reviewed daily for completeness. The questionnaire contained different variables such as sociodemographic and economic characteristics, clinical and anthropometric information.

## Antiretroviral therapy adherence

Antiretroviral treatment adherence is the level to which a person is taking medicine as prescribed by a physician and as per medical recommendations in relation to timing, dosing and consistency, and correctly taking the drugs in terms of right doses and right times (*Chaiyachati et al., 2014*). In this study, patient self-assessments were collected using the visual analogue scale (VAS) to measure primary outcome and adherence. VAS method was used to assess the adherence mostly in resource-limited settings and the most effective and feasible method available at the time the study, and shown to predict virological response in clinical trials and clinical routines (*Finitsis et al., 2016*; *Chkhartishvili et al., 2014*). With the VAS approach, a series of scales with values ranging from 0 to 100% is used to assess adherence during the past 30 days (*Chesney et al., 2000*). The participants were asked to place an "X" inside the box above the point showing the best guess about how

much of their current antiretroviral medications were taken in the past 30 days. Adherence at a VAS score of ≥ 95% over the past 30 days was defined as adherent and less than 95% as non-adherent.

## Outcome measures

To investigate the extent and factors associated with non-adherence to ART among people living with HIV and on treatment, we used socioeconomic indicators such as education, income, asset possession, employment status, dietary diversity, nutritional status and household food insecurity. Sociodemographic characteristics such as age, gender and marital status; clinical characteristics such as CD4 count, duration on ART and history of opportunistic infections were taken into consideration during the interviews and chart review for data collection. Inclusion of independent variables were based on literature reviews conducted, data availability, and theoretical relevance (*Tabachnick & Fidell, 2007*). We chose these independent variables because of extensive literature highlighting the relationship between these covariates and ART adherence (*Fonsah et al., 2017*; *Eyassu, Mothiba & Mbambo-Kekana, 2016*; *Alagaw et al., 2013*; *Peltzer & Pengpid, 2013*; *Bangsberg, 2006*).

Food insecurity is defined as the economic and social condition of limited or uncertain access to adequate food (*US Department of Agriculture & Economic Research Service, 2019*). The access component of household food insecurity was measured using the standard household food insecurity access scale (HFIAS) (*Coates, Swindale & Bilinsky, 2007*). This is a nine-item questionnaire assessing household food insecurity in the domains of anxiety about household food access, insufficient quality of food and insufficient food intake in the past 30 days. In this study, HFIAS was calculated by summing the score for all nine items and ranges from 0 to a maximum of 27. A higher HFIAS score indicates poor access to food and greater food insecurity. Respondents were assigned to discrete categories of food insecurity severity: (1) food secure, (2) mildly food insecure, (3) moderately food insecure, and (4) severely food insecure, which we dichotomized into food insecure versus food secure. The alpha value for internal consistency of the scale was 0.91.

Malnutrition is the condition that occurs when the body does not get enough nutrients (*US National Library of Medicine, 2019*). In this study, malnutrition refers to undernutrition. The researchers used anthropometric measurement such as weight and height to measure malnutrition. The participants' weight was measured by means of the Seka weight scale calibrated to the nearest 0.1 kg after removing heavy clothes. The participants' height was measured using the Seka measuring rod calibrated to the nearest 0.1 cm. The participants took off their shoes, stood erect and looked straight in the horizontal plane to measure their height. The body mass index (BMI) was calculated as weight in kilograms divided by the square of height in meters ($kg/m^2$). Participants with a BMI of less than 18.5 $kg/m^2$ were considered malnourished.

Household dietary diversity is the economic ability of a household to access a variety of foods during the past seven days. Household dietary diversity was measured based on the dietary measurement method (*Kennedy, Ballard & Dop, 2011*; *Swindale & Bilinksy, 2006*). Twelve questions were used to assess dietary diversity. Participants were asked to report the
frequency of consumption of each of the following 12 food groups: (1) cereals; (2) roots and tubers; (3) pulses and legumes/nuts; (4) vegetables; (5) fruits; (6) meat and poultry; (7) eggs; (8) fish and seafood; (9) milk and milk products; (10) oils and fats; (11) sugar and sweets; and (12) miscellaneous. Irrespective of the frequency, participants received '1' point if they consumed at least once during the last seven days of the foods within each subgroup, and '0' points if they never consumed the food. The household dietary diversity score (HDDS) was constructed as the sum of some food groups consumed over the past week, ranging from 0 to 12. A high value indicated a diversified diet. The mean household dietary diversity score in the study subjects was categorized as adequate if the dietary diversity score is nine or above, and inadequate, when the diversity score is below nine.

Asset possession refers to a household's possession of assets elicited by asking participants a series of 13 questions about household assets and housing characteristics such as housing quality (floor, walls and roof material), source of drinking water, type of toilet facility, presence of electricity, type of cooking fuel, and ownership of modern household durable goods and livestock (e.g., bicycle, television, radio, motorcycle, telephone, refrigerator, mattress, bed and mobile phone). Following the method of Filmer and Pritchett (*Filmer & Pritchett, 2001*), principal components analysis was applied to define the asset wealth index among the PLWHA, and based on wealth index score participants were sorted into three quintiles of relative asset lower, middle and upper. A high value indicates more asset possession.

## Data analyses

To assess factors associated with non-adherence to ART, binary adherence variables using data from VAS was created. Both descriptive and analytic statistics were used. Binary logistic regression analysis was used to find associations between the independent variables and the outcome variable. The Chi-square test of independence was used to find factors for not taking medication. All the variables showing a significant association in a binary analysis at $p < 0.25$ were entered to multivariable logistic regression to identify factors which have statistically significant association. In the final multivariate analysis, the test was two-sided and $p$-value $<0.05$ was considered statistically significant. All statistical analyses were conducted in IBM SPSS Statistics for Windows, Version 24.0 (IBM Corporation, Armonk, NY, USA).

## Ethical approval and consent to participate

Before the study begins, ethical clearance was obtained from the Health Studies Higher Degrees Committee of the College of Human Sciences at the University of South Africa (UNISA) (REC 012714-039 NHERC). Further, permission to conduct the study (ethical clearance) was sought after authorised cooperation letter was written from UNISA Ethiopia regional learning centre to Benishangul Gumuz Regional State Health Bureau (BG-RHB) and thus obtained the ethical clearance. A support letter to conduct the study among the selected health facilities in the region was further written to those health facilities by BGR-RHB. Institutional consent and permission to conduct an interview with PLWHA among

the selected health facilities were sought. Authorities at the study sites were assured that any information that the researcher came across during the conduct of the research was not be disclosed to any interest groups that could jeopardize the participants' welfare in society and the concerned institution. The researcher abided the guideline of the institutions, which has the right to terminate the study if the safety and confidentiality of participants might be compromised. All participants provided written informed consent to indicate their voluntary participation in the study before administering any instrument. The researchers ensured that participants understood their rights of voluntary participation, anonymity and confidentiality. ART follow-up records were retrieved by the health providers working in the ART units, and personal information of the participants was kept confidential.

## RESULTS

Three hundred and ninety-four study participants were recruited for this study. Only four participants declined or terminated to participate into the study. The majority (80%) of the participants were aged between 25 and 44 years. The women participants constituted (66.4%) of the total participants. Nearly (50%) of the participants were married and an equal number of the participants were divorced, widowed or single. About (36.4%) of them never attended school, while only (4%) attended college/university. The majority of the participants (91%) lived in urban areas. Table 2 shows the characteristics of the study participants.

In terms of socioeconomic characteristics, most of the participants were poor as measured by the wealth index, a proxy measure for asset possession. About (60%) of them were in the lower asset tertiles. More than a quarter of them were unemployed (31%). A large number of the participants (67.7%) lived on a poor mean monthly income of 1,260 Ethiopian Birr (equivalent to US$45), below the World Bank poverty threshold of US$ 1.90/day (*The World Bank Group, 2019*). The vast majority of the study participants (76%) were food insecure and (60%) had a BMI of less than 18.5 kg/m$^2$. The mean CD4 count was 559 (SD = 319.6) cells/mm$^3$, with a range of 60 to 1,914 cell/mm$^3$. More than a quarter of them (33.6%) suffered from frequent opportunistic infection in the last three months.

Based on a 30-day adherence measurement, (39.7%) of the study participants were found non-adherent to ART. Of the food insecure participants, more than half (51.4%) was non-adherent; while (47.2%) of malnourished PLWHA, those whose BMI is below 18.5 kg/m$^2$ was non-adherent to ART.

### Factors associated with non-adherence to antiretroviral therapy

As presented in Table 3, in the binary logistic regression analysis age below 25 years, marital status, urban residential area, never been to school, lack of employment, food insecurity, BMI below 18.5 kg/m$^2$, household dietary diversity, poor asset possession, CD4 cell count below 350 cell/mm$^3$, and history of opportunistic infections were found associated with non-adherence to ART. But subsequent multivariate logistic regression analysis showed age below 25 years, lack of employment, food insecurity, malnutrition and opportunistic infections were among the factors associated with the factors in non-adherence to ART.

Nigusso and Mavhandu-Mudzusi (2020), *PeerJ*, DOI 10.7717/peerj.8558

**Table 2** Characteristics of study participants, $N = 390$.

| Characteristics | Non-adherent to ART ($n = 155$) | Adherent to ART ($n = 235$) |
|---|---|---|
| *Sociodemographic characteristics* | | |
| Age | | |
| Less than 25 years | 19 (12.0) | 14 (6.0) |
| 25–35 years | 76 (49.0) | 106 (45.1) |
| More than 35 years | 60 (38.7) | 115 (48.9) |
| Gender: | | |
| Female | 110 (71.0) | 149 (63.4) |
| Male | 45 (29.0) | 86 (36.6) |
| Marital status: | | |
| Divorced/Widowed/Single | 67 (43.2) | 128 (54.5) |
| Married | 88 (56.8) | 107 (45.5) |
| Religious: | | |
| Christian | 117 (75.5) | 184 (78.3) |
| Muslim | 38 (24.5) | 51 (21.7) |
| Residence area: | | |
| Urban | 149 (96.1) | 206 (87.7) |
| Rural | 6 (3.9) | 29 (12.3) |
| *Socioeconomic factors* | | |
| Education level: | | |
| Never been to school | 57 (36.8) | 85 (36.2) |
| Primary level | 69 (44.5) | 97 (41.3) |
| Secondary level | 27 (17.4) | 39 (16.6) |
| College/University level | 2 (1.3) | 14 (6.0) |
| Employment status: | | |
| Unemployed | 70 (45.2) | 52 (22.1) |
| Employed | 85 (54.8) | 183 (77.9) |
| Monthly income (in Ethiopian Birr) | | |
| <1750 | 119 (76.8) | 174 (74) |
| ≥1750 | 36 (23.2) | 61 (26) |
| Food security status: | | |
| Food insecure | 152 (98.1) | 144 (61.3) |
| Food secure | 3 (1.9) | 91 (38.7) |
| BMI score (in kg/m$^2$): | | |
| <18.5 | 111 (71.6) | 124 (52.8) |
| ≥18.5 | 44 (28.4) | 111 (47.2) |
| Household dietary diversity: | | |
| Inadequate | 121 (78.1) | 139 (59.1) |
| Adequate | 34 (21.9) | 96 (40.9) |

| Characteristics | Non-adherent to ART (*n* = 155) | Adherent to ART (*n* = 235) |
|---|---|---|
| Household wealth index score | | |
| Lower | 76 (49.0) | 158 (67.2) |
| Middle | 35 (22.6) | 41 (17.4) |
| Upper | 44 (28.4) | 36 (15.3) |
| *Clinical features* | | |
| Duration of ART initiation: | | |
| Less than 12 months | 13 (8.4) | 24 (10.2) |
| 1–5 years | 75 (48.4) | 80 (34.0) |
| 5–10 years | 56 (36.1) | 112 (47.7) |
| >10 years | 11 (7.1) | 19 (8.1) |
| CD4 count (in Cell/mm$^3$): | | |
| <350 | 54 (34.8) | 54 (23.0) |
| 350–500 | 26 (16.8) | 55 (23.4) |
| >501 | 75 948.4) | 126 (53.6) |
| History of opportunistic infections in the last three months: | | |
| Yes | 76 (49.0) | 55 (23.4) |
| No | 79 (51.0) | 180 (76.6) |

**Notes.**

n, frequency in number; %, percentage.

Age groups of less than 25 years were found to be four-fold non-adherent to ART in comparison with older age groups (AOR: 4.30, 95% CI [1.39–3.35]; $p = 0.011$). Urban residential area is also found contributing for non-adherence to ART (AOR: 2.78, CI [1.23–7.09], $p = 0.043$). The odds of non-adherence to ART among unemployed PLWHA were almost twice higher than their peers who were employed (AOR: 1.75, 95% CI [1.05–2.91]; $p = 0.032$). Food insecurity was the other socioeconomic factor strongly found impeding ART adherence (AOR: 2.67, 95% CI [7.59–8.97]; $p < 0.0001$), and malnutrition was among factors deterring PLWHA to attain the required $\geq$ 95% ART adherence level (AOR: 1.55, 95% CI [1.94–2.56]; $p = 0.027$). Recurrent episodes of opportunistic infection (AOR: 1.81, 95% CI [1.11–2.97]; $p = 0.018$) was a clinical factor contributing to non-adherence to ART.

## Reasons for missing medication

The prevalence of non-adherence to ART in this study was found to be (39.7%). Among these, (83%) of them missed twice a month while (4.4%) missed three times or more a month. Four reasons were reported by the study participants as important reasons for missing their medications: (i) demanding work or household responsibilities (33%), (ii) ran out of medication (34%), (iii) unable to take without food (22%), and (iv) forgetfulness (11%). Table 4 shows self-reported reasons for missing medications.

**Table 3  Bivariable and multivariable analysis of factors associated with non-adherence to ART among peoples living with HIV and AIDS attending antiretroviral therapy in Benishangul Gumuz Region, Ethiopia, 2019.**

| Characteristics | Bivariable regression analysis | | | Multivariable regression analysis | | |
|---|---|---|---|---|---|---|
| | COR | 95% CI | p-value | AOR | 95% CI | p-value |
| *Sociodemographic characteristics* | | | | | | |
| Age | | | | | | |
| Less than 25 years | 2.6 | [1.22, 5.55] | 0.013 | 4.30 | [1.39, 3.35] | 0.011 |
| 25–35 years | 1.4 | [0.90, 2.10] | 0.157 | 1.23 | [0.75, 2.02] | 0.411 |
| More than 35 years | Ref | | | | | |
| Gender: | | | | | | |
| Female | 0.70 | [0.46, 1.10] | 0.120 | 1.09 | [0.63. 1.09] | 0.67 |
| Male | Ref | | | | | |
| Marital status: | | | | | | |
| Divorced/Widowed/Single | 1.60 | [1.04, 2.36] | 0.030 | 0.63 | [0.39,1.01] | 0.054 |
| Married | Ref | | | | | |
| Religious: | | | | | | |
| Christian | 1.17 | [0.71, 1.90] | 0.520 | | | |
| Muslim | Ref | | | | | |
| Residence area: | | | | | | |
| Urban | 3.50 | [1.42, 8.60] | 0.007 | 2.78 | [1.23, 7.09] | 0.043 |
| Rural | Ref | | | | | |
| *Socioeconomic factors* | | | | | | |
| Education level: | | | | | | |
| Never been to school | 4.70 | [1.03, 21.0] | 0.046 | 1.07 | [0.17, 6.71] | 0.38 |
| Primary level | 4.90 | [1.1, 22.0] | 0.038 | 1.44 | [0.23, 0.89] | 0.63 |
| Secondary level | 4.80 | [1.02, 23] | 0.047 | 1.42 | [0.21, 9.46] | 0.71 |
| College/University level | Ref | | | | | |
| Employment status: | | | | | | |
| Unemployed | 2.89 | [1.86, 4.50] | <0.001 | 1.75 | [1.05, 2.91] | 0.032 |
| Employed | Ref | | | | | |
| Monthly income (in Ethiopian Birr) | | | | | | |
| <1,750 | 0.86 | [0.54, 1.39] | 0.540 | | | |
| ≥1,750 | Ref | | | | | |
| Food security status: | | | | | | |
| Food insecure | 3.2 | [9.9, 11.3] | <0.0001 | 2.67 | [7.59, 8.97] | <0.0001 |
| Food secure | Ref | | | | | |
| BMI score (in kg/m$^2$): | | | | | | |
| <18.5 | 2.3 | [2.47, 4.48] | <0.0001 | 1.55 | [1.94, 2.56] | 0.027 |
| ≥18.5 | Ref | | | | | |
| Household dietary diversity: | | | | | | |
| Inadequate | 2.5 | [1.60, 3.90] | <0.0001 | 0.93 | [0.51, 1.68] | 0.81 |
| Adequate | Ref | | | | | |

**Table 3** (*continued*)

| Characteristics | Bivariable regression analysis | | | Multivariable regression analysis | | |
|---|---|---|---|---|---|---|
| | COR | 95% CI | p-value | AOR | 95% CI | p-value |
| Household wealth index score | | | | | | |
| Lower | 1.8 | [1.05, 3.01] | 0.03 | 0.63 | [0.33, 1.13] | 0.12 |
| Middle | 2.5 | [1.51, 4.27] | 0.00 | 0.72 | [0.35, 1,47] | 0.36 |
| Upper | Ref | | | | | |
| *Clinical features* | | | | | | |
| Duration of ART initiation: | | | | | | |
| Less than 12 months | 0.94 | [0.34, 2.50] | 0.890 | | | |
| 1–5 years | 1.60 | [0.72, 3.60] | 0.240 | | | |
| 5–10 years | 0.86 | [0.39, 1.90] | 0.720 | | | |
| >10 years | Ref | | | | | |
| CD4 count (in Cell/mm$^3$): | | | | | | |
| <350 | 1.68 | [1.10, 2.60] | 0.032 | 1.1 | [0.57, 1,78] | 0.92 |
| 350–500 | 0.79 | [0.46, 1.40] | 0.410 | 0.77 | [0.39, 1.46] | 0.42 |
| >501 | Ref | | | | | |
| History of opportunistic infections in the last three months: | | | | | | |
| Yes | 3.20 | [2.10, 4.90] | <0.001 | 1.81 | [1.11, 2.97] | 0.018 |
| No | Ref | | | | | |

**Notes.**

COR, crude odds ratio; AOR, adjusted odds ratio; OR, odds ratio; CI, confidence interval; Ref, reference.

**Table 4** Self-reported reason for missing ART among peoples living with HIV and AIDS attending antiretroviral therapy in Benishangul Gumuz Region, Ethiopia, 2019.

| Characteristics | Frequency | Percent (%) | $\chi^2$ | p-value |
|---|---|---|---|---|
| ART adherence: | | | | |
| Adherent | 235 | 60.3 | | |
| Non-adherent | 135 | 39.7 | | |
| Reasons for not taking medication in the last seven days: | | | | |
| Demanding work or household responsibilities | 44 | 33.0 | | |
| Ran out of medication | 46 | 34.0 | 3.01 | 0.009 |
| Unable to take without food | 30 | 22.0 | | |
| Forgetfulness | 15 | 11.0 | | |
| Frequency of missing in the last seven days: | | | | |
| Two times | 112 | 83.0 | 5.48 | 0.043 |
| Three times | 17 | 12.6 | | |
| Above three | 6 | 4.4 | | |
| VAS adherence score: | | | | |
| $M = 86.40$ | | | | |
| $SD = 14.94$ | | | | |

**Notes.**

$\chi^2$, Chi-square test; VAS, Visual Analogue Scale.

## DISCUSSION

In the current study, the factors associated with non-adherence to ART among PLWHA in Benishangul-Gumuz were examined. Our findings endorse higher prevalence of ART non-adherence, with only 60.3% of our study participants remaining adherent to ART signposts the requirements of more efforts towards commitment to and realization of a 90-90-90 HIV treatment initiative. The study found age groups below 25 years were non-adherent to ART. Analogous to this finding, the results of a meta-analysis study (*Ghidei et al., 2013*) showed a higher risk of non-adherence among young PLWHA while older PLWHA had reduced risk of non-adherence to ART. Another study in Canada found a higher proportion of non-adherence among young PLWHA (*Hadland et al., 2012*). Youths are willing to engage in greater risky sexual behaviors and may also be more likely to assume the risks of poor HIV medication compliance, thus youth tailored behavioral change interventions that address the special emotional and developmental concerns of younger PLWHA could be designed.

Strong plausible evidence of the association between non-adherence to ART and food insecurity was found. Food insecure PLWHA were found twice more likely non-adherence to ART than their food secure counterparts. Studies in both developing and developed countries have culminated the strong association between non-adherence to ART and food insecurity effects. For example, studies in Lake Victoria,Kenya (*Nagata et al., 2012*) and in Atlanta, USA (*Pellowski et al., 2016*) showed that food insecure PLWHA miss doses of their daily medication. Similarly, studies in Jimma, southwest Ethiopia (*Tiyou et al., 2012*), and Congo (*Musumari et al., 2014*), and established a strong association between non-adherence to ART and food insecurity. Since non-adherence to ART can risk developing viral resistance (*Weiser et al., 2012*; *Musumari et al., 2014*) and food insecurity poses a significant challenge to the HIV/AIDS response (*Pellowski et al., 2016*; *Federal Ministry of Health of Ethiopia, 2015*; *Kalichman et al., 2014*).

In similar with the findings of other studies in Ethiopia (*Berhe, Tegabu & Alemayehu, 2013*), the current study found malnourished PLWHA were non-adherent to ART. This is also reported in another matched case-control study in the Central Tigray Zone, Ethiopia that revealed malnutrition was among the independent factors associated with lack of adherence to ART (*Weldehaweria et al., 2017*). Absence of nutritional support fuels the synergistic co-existence of HIV and malnutrition that has a perverse effect on ART adherence and the wellbeing of PLWHA (*Mekuria et al., 2015*; *Sicotte et al., 2014*; *Berhe, Tegabu & Alemayehu, 2013*). Therefore, nutritious food support is highly regarded as a viable option to increase weight gain (*Audain et al., 2015*) and improve treatment outcome (*Chaiyachati et al., 2014*; *Tiyou et al., 2012*).

Our findings are similar to the study by Nachega et al., who found a strong association between non-adherence to ART and lack of employment among PLWHA living both in low-income and in high-income countries (*Nagata et al., 2012*). In opposition to this finding, unemployed PLWHA in Nigeria had better ART adherence than employed people and rarely missed their ART medications (*Okoronkwo et al., 2013*). The possible explanation for adherence among unemployed PLWHA might be time freedom to take

their medications without forgetting being a concern. Also, they are more privileged to take their medications freely without any psychological impacts that put them under stress such as the workplace environment (*Okoronkwo et al., 2013*). Education status, dietary diversity and asset possession were among the alterable socioeconomic factors found influencing adherence to ART, thus developing effective adherence interventions to such alterable barriers are important in achieving an optimum adherence level.

Opportunistic infections were found to affect the treatment outcome of PLWHA. According to the results of this study, those who suffered recurrent illness within the last three months were twice more likely to miss their treatment and be non-adherent to ART. This finding is consistent with the findings of other studies that reported the presence of a significant association between treatment for HIV and other infections and adherence to ART (*Eyassu, Mothiba & Mbambo-Kekana, 2016*). Corresponding to this finding, non-adherence to ART as a result of opportunistic infection was reported among PLWHA in Yaounde, Cameroon (*Fonsah et al., 2017*) and the Republic of Korea (*Kim et al., 2018*). The reason for non-adherence to ART among sick PLWHA include the pill burden and poor absorption (*Fonsah et al., 2017*; *Eyassu, Mothiba & Mbambo-Kekana, 2016*).

Unlike the study of *Schafer et al. (2017)*, the findings of the current study reveals urban residential area as a risk factor for non-adherence to ART. This findings are consistent with the evidence that urban residences, as a result of dynamic socioeconomic alteration, may lack the desired adherence level (*Belayihun & Negus, 2015*). Similar to this finding, a study at Gonder University in northwest Ethiopia revealed that urban residential areas are associated with factors of non-adherence to ART (*Molla et al., 2018*). To alleviate the adherence issues both in urban and rural setups, a robust and sustainable response has to be foresighted, including using community volunteers to re-engage patients in care and helping them to remain in care, and empowering PLWHA in rural areas to regularly visit their healthcare provider (*Kaihin et al., 2015*; *Campbell et al., 2012*).

The current study has several limitations: first, even if an attempt was made to minimize social desirability bias, it might be subjected to recall and social desirability biases as ART adherence, food security and dietary diversity were assessed based on the self-reported measure. Second, the study design was cross-sectional in nature and hence cannot be used to determine causality. Malnutrition can be both a cause and effect of non-adherence to ART. Hence, future longitudinal studies are warranted to ascertain the causal directions of these variables.

## CONCLUSIONS

The current study provides empirical findings on the factors associated with non-adherence to ART and the extent of non-adherence in Benishangul-Gumuz. The magnitude of non-adherence is as high as (39.7%) in the region. High prevalence of non-adherence is strongly associated with young age below 25 years, urban residential area, lack of employment, food insecurity, malnutrition and opportunistic infections. Therefore, improving ART adherence requires the adoption of innovative youth-centered adherence enhancing strategies, creation of employment opportunities, devising follow-up and mitigation of

recurrent opportunistic illness, and integrating nutrition and food security interventions with the HIV/AIDS program that can be implemented through the multisectoral approach to bridge the resource gap and improve the treatment outcome across all setups.

**List of abbreviations**

| | |
|---|---|
| **AIDS** | Acquired Immune Deficiency Syndrome |
| **ART** | Antiretroviral Treatment |
| **BG-RHB** | Benishangul-Gumuz Regional State Health Bureau |
| **BMI** | Body Mass Index |
| **HDDS** | Household Dietary Diversity score |
| **HFIAS** | Household Food Insecurity Access Scale score |
| **HIV** | Human Immunodeficiency Virus |
| **PLWHA** | people living with HIV and AIDS |
| **UNISA** | University of South Africa |
| **WFP** | United Nations World Food Programme |

## ACKNOWLEDGEMENTS

We are grateful to all the research assistants, the data entry clerks and the PLWHA who voluntarily participated in the survey; and to the University of South Africa for the support provided to us.

### Funding
The authors received no funding for this work.

### Competing Interests
The authors declare there are no competing interests.

### Author Contributions

- Fikadu Tadesse Nigusso conceived and designed the experiments, performed the experiments, analyzed the data, prepared figures and/or tables, authored or reviewed drafts of the paper, and approved the final draft.
- Azwihangwisi Helen Mavhandu-Mudzusi performed the experiments, authored or reviewed drafts of the paper, and approved the final draft.

### Human Ethics

The following information was supplied relating to ethical approvals (i.e., approving body and any reference numbers):

Health Studies Higher Degrees Committee of the College of Human Sciences at the University of South Africa approved the study (Ref: REC 012714-039 NHERC).

## Ethics

The following information was supplied relating to ethical approvals (i.e., approving body and any reference numbers):

Health Studies Higher Degrees Committee of the College of Human Sciences at the University of South Africa approved the study (Ref: REC 012714-039 NHERC).

## Data Availability

Raw data is available as a Supplemental File.

## Supplemental Information

Supplemental information for this article can be found online at http://dx.doi.org/10.7717/peerj.8558#supplemental-information.

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
