# Peer review of "Magnitude of non-adherence to antiretroviral therapy and associated factors among adult people living with HIV/AIDS in Benishangul-Gumuz Regional State, Ethiopia"

_PeerJ, doi:10.7717/peerj.8558_

## Round 0.1 · original submission · Major Revisions

Please address all comments. Please also include information about receipt of ethics approval for this study from Ethiopia research regulatory agencies.

·

Basic reporting

comments:
a) Title:
-better to make it more self-explanatory
-it should reflect the main aim of the study in unambiguous ways..
-it lacks clarity and seems a bit uneasy and
- hence it should be corrected as " Treatment adherence and associated factors among adult PLWHA......."
b) Author's address: should be complete (e.g; P.O.box, phone,emails etc.
c) grammatical errors and idea flow:
- some statements do not give sense (e.g;under introduction line No.69& 70 should be re-written in grammatical order,line No.80 referencing style, in result part line No.229 lacks unit of measurement,line No.251-57 incorrect statement,
-result presentation in number and % is not uniform (see comments)

Experimental design

comments on Methods and Participants
-Methods employed are not described with sufficient information
-The knowledge gap being investigated has not been adequately identified
-Sampling procedure (allocation) is unclear and not convincing (it should be depicted graphically how the sample is drawn)
-the investigation does not seem rigorous (if not sufficient justification should be given)
-inclusion-exclusion criteria lacks clarity
-conceptual frame work is not provided for variables
- findings of pre-test should be explained if any
-lacks data quality managements methods used
-justification should be given for selection of adherence assessment methods(since there are many methods) ...why VAS method??? its validity ??? applicability in clinical practice?
-Issue of confidentiality is not adequately addressed
-specific objectives should be clearly stated
otherwise the study is good.

Validity of the findings

-findings should reflect the set specific objectives(not stated) of the study
-inconclusive/inconsistence findings have not been adequately discussed
and please,check aforementioned comments.

Additional comments

General comments:
The following things have not been adequately addressed or no justification given...
a) patients with co-morbid conditions (TB,CHF,HTN,CKD,DM etc...??)
b) data on history of substance use(alcohol or tobacco use vs adherence??)
c) drug related factors (data on concurrently used medications, type of regimen used, if any(drug-drug interaction, ADR etc..??)
d) duration of illness vs adherence ?????
e) length of ART use (in years) vs adherence ?????
f) Hx of hospitalization while on treatment ???
g) health care system related factors (drug availability, counseling service, care provider ????etc...)
h) research question should be clearly stated
Otherwise,the study is legible, of current government and community concern and has clearly forwarded recommendations.
Finally, I appreciate the authors for their effort and commitment and I assuredly tell you that my constructive comments will further enhance the quality of your study. And hence please,go through all my concerns and improve accordingly.
Kind regards!

Reviewer 2 ·

Basic reporting

No comments

Experimental design

no comments

Validity of the findings

No comments

Additional comments

This manuscript titled "Adherence to antiretroviral therapy among people living with HIV in Benishangul-Gumuz Regional State, Ethiopia" is very well drafted and written. However, there is no novelty in this paper, but maybe it will help in planning future antiviral therapy program in Gumuz Regional State, Ethiopia

Reviewer 3 ·

Basic reporting

no comment

Experimental design

no comment

Validity of the findings

no comment

Additional comments

Comments
Abstract
1. In the last sentence of the introduction section of your abstract (from line 24-26) you showed your objective that is good. But my comment is regarding your title. Here you want to show the prevalence of non-adherence to ART. So better to modify your title” Magnitude of non-adherence to ART treatment and associated factors among people living with HIV in Benishangul-Gumuz Regional State,2 northwest Ethiopia”
2. Methods section
Line 27: Do you mean all sampled population participated in the study? Your response rate was 98.9 % which means from your sampled population around 389 participants were participated. So try to see it clarity
3. Conclusion: First conclude about the outcome variable then you can recommend for your outcome or independent variables. You try to mention the recommendation but better to put recommendation as a title with conclusion.
Methods
1. Line 119-120: you exclude patients with cognitive problem,i.e. good but my concern is How do you know this population to exclude? Were you use tool to screen or what?

Results
1. Line 230:……. >95% (what is 95%? It is not clear,is that to mean confidence? And correct though out your data)
2. Line 242-243: ……..other socioeconomic factor strongly found impeding ART adherence (AOR: 2.1, CI: 6.1–56, p <0.001) .The confidence interval is too large. This might be because of small number of patients who have food insecure and that is why your confidence interval is high. The other thing is you measured food insecurity by HFIAS as continuous variable. Your model was logistic regression. How you manage/enter to SPSS both categorical and continues variable simultaneously and how you used AOR for continuous variable to explain risk? As we can see here majority of participants were near to zero i.e. below mean. So better to clear with your analysis and description.If you have reference/possibility better to dichotomize HFIAS rather than taking as continuous variable.
3. found impeding ART adherence (AOR: 2.1, CI: 6.1–56, p <0.001) and malnutrition was among factors deterring PLWHA to attain the required >95% ART adherence level (AOR: 0.9, CI:0.82–0.98, p = 0.018).This AOR shows being malnourished is protective? this is contrary to a logic even difficult for recommendation. So better to have a look the analysis.
4. Line 282-283 and 321-323: recommendation is better to be at the end with conclusion
5. Table 2: 1-In your table add frequencies for each variable since we can not be sure whether your COR is correct or not
2- show the AOR some variables like gender,Religion......I know this variables were not eligible for multivariate but better to show its AOR with its CI or if you were taking bivariable, no need of listing this variables here

6.Regarding ethical consideration there is no information about the study from the regional health biro or any concerned body from Ethiopia. Add informations who allow the study from the region health biro. To whom you summit the ethical clearance after you got from College of Human Sciences at the University of South Africa.

Annotated reviews are not available for download in order to protect the identity of reviewers who chose to remain anonymous.

---

## Round 0.2 · Major Revisions

Thanks for submitting your manuscript. I look forward to your prompt review of the manuscript. Feel free to ask for clarification where there is a need

Reviewer 3 ·

Basic reporting

No

Experimental design

No

Validity of the findings

No

Additional comments

Comments to the authors
Abstract
Methods section
1. Line 27: Do you mean all sampled population participated in the study? Your response rate was 98.9 % which means from your sampled population around 390 participants were gave a response. Still it lacks some clarity. “We studied 394 peoples living…..” For me all 394 participants gave a response. So better to mention with only with 98.9% response rate rather than mentioning the frequency (394) to avoid the confusion. Then avoid “s “from people…….”
Conclusion:
2. How do you conclude about magnitude of non-adherence here? What factors influence non adherence? But most of the statements seems recommendation rather than conclusion. So either add recommendation with conclusion as a title or modify your conclusion. Even some contents seems out of your objective.
Methods
1. Line 125-126: you exclude patients with cognitive problem, i.e. well but my concern is how do you know this population to exclude? Were you use tool to screen or what?
Were you exclude known mental health problem or were you screening during interview?
If you were screening during interview your data collectors were not appropriate to screen all mental health problems. But if already patients were known and diagnosed before that is possible. Otherwise you needed to use screening tool.
2. Line 279-280: ……..other socioeconomic factor strongly found impeding ART adherence (AOR: 2.48, 95% CI: 11.3, 3.2; p <0.0001). You tried to manage large confidence interval but sill reversed. So have a look at again.
3. Table 2: 1-In your table add frequencies for each variable since we cannot be sure whether your COR is correct or not. Yes off course, I see the frequencies but I want to say the crosstab frequencies. For example gender (female (non-adherence/yes---?, no--?) ….male non adherence( yes-----?,No…..?) and your table needs modification. It is not necessary to put p-value of COR.
4. 2- show the AOR some variables like gender, religion......I know these variables were not eligible for multivariate but better to show its AOR with its CI. I know this variables are not associated at multivariate but it fulfil the minimum criteria. So you need to show further this variables is not associated at multivariate with its AOR and CI.
5. “Permission to conduct the study was obtained from the Benishangul Gumuz Regional State Health Bureau. The respective health facilities gave permission prior to data collection” In what way you get permission from Health Bureau written or verbal? Because you said twice permission i.e health facility and Health Bureau. So state what type of permission you got.

Annotated reviews are not available for download in order to protect the identity of reviewers who chose to remain anonymous.

---

## Round 0.3 · Minor Revisions

Thanks for submitting your important work to the journal

Please improve the abstract - in the methodology section of the abstract, please include information on the explanatory and outcome variables studied, I look forward to your prompt response to enable me move the manuscript to publication this year.

Reviewer 2 ·

Basic reporting

This manuscript titled "Adherence to antiretroviral therapy among people living with HIV in Benishangul-Gumuz Regional State, Ethiopia" is very well drafted and written. However, there is no novelty in this paper, but maybe it will help in planning future antiviral therapy program in Gumuz Regional State, Ethiopia

Experimental design

No comment

Validity of the findings

No comment

Additional comments

This manuscript titled "Magnitude of non-adherence to antiretroviral
therapy and associated factors among adult people
living with HIV/AIDS in Benishangul-Gumuz Regional
State, Ethiopia" is a resubmission of the previous article "Adherence to antiretroviral therapy among people living with HIV in Benishangul-Gumuz Regional State, Ethiopia" has been very well-drafted and written. The authors have corrected most of the issues raised by other reviewers. This paper will help in planning future antiviral therapy program in Gumuz Regional State, Ethiopia. I would recommend this paper for publication.

Reviewer 3 ·

Basic reporting

no

Experimental design

no

Validity of the findings

no

Additional comments

I am satisfied with your respones

---

## Round 0.4 · Minor Revisions

Dear Dr Nigusso,

Please include the details of the variables. You wrote - Sociodemographic and economic indicators, and clinical characteristics were explanatory variables. It is important that the specific details of what were the sociodemographic variables, economic variables and clinic characteristics be listed so the result has meaning. If you are concerned with word count, please reduce the introduction to 2 sentences. Introduction is too long. Looking forward to the prompt revision. Happy 2020

---

## Round 0.5 · accepted · Accept

Congratulations. Thanks for going through the requested revisions. I look forward to seeing how the manuscript makes a difference to the field of HIV patients' care